# Comprehensive Insights into the Remarkable Function and Regulatory Mechanism of FluG during Asexual Development in *Beauveria bassiana*

**DOI:** 10.3390/ijms25116261

**Published:** 2024-06-06

**Authors:** Fang Li, Juefeng Zhang, Haiying Zhong, Kaili Yu, Jianming Chen

**Affiliations:** Institute of Plant Protection and Microbiology, Zhejiang Academy of Agricultural Sciences, Hangzhou 310021, China; lifang870910@163.com (F.L.); zhy8085@163.com (H.Z.); yukaili17@zju.edu.cn (K.Y.)

**Keywords:** *Beauveria bassiana*, asexual development, upstream developmental regulatory protein, signaling pathway, protein interaction, virulence

## Abstract

Asexual development is the main propagation and transmission mode of *Beauveria bassiana* and the basis of its pathogenicity. The regulation mechanism of conidiation and the key gene resources for utilization are key links to improving the conidia yield and quality of *Beauveria bassiana*. Their clarification may promote the industrialization of fungal pesticides. Here, we compared the regulation of morphology, resistance to external stress, virulence, and nutrient utilization capacity between the upstream developmental regulatory gene *fluG* and the key genes *brlA*, *abaA*, and *wetA* in the central growth and development pathway. The results showed that the Δ*brlA* and Δ*abaA* mutants completely lost the capacity to conidiate and that the Δ*wetA* mutant had seriously reduced conidiation capacity. Although the deletion of *fluG* did not reduce the conidiation ability as much as deletions of *brlA*, *abaA*, and *wetA*, it significantly reduced the fungal response to external stress, virulence, and nutrient utilization, while the deletion of the three other genes had little effect. Via transcriptome analysis and screening the yeast nuclear system library, we found that the differentially expressed genes in the Δ*fluG* mutants were concentrated in the signaling pathways of ABC transporters, propionate metabolism, tryptophan metabolism, DNA replication, mismatch repair, and fatty acid metabolism. FluG directly acted on 40 proteins that were involved in various signaling pathways such as metabolism, oxidative stress, and cell homeostasis. The analysis indicated that the regulatory function of *fluG* was mainly involved in DNA replication, cell homeostasis, fungal growth and metabolism, and the response to external stress. Our results revealed the biological function of *fluG* in asexual development and the responses to several environmental stresses as well as its influence on the asexual development regulatory network in *B. bassiana*.

## 1. Introduction

Among entomopathogenic fungi, *Beauveria bassiana* is a main source of the broadest-spectrum fungal pesticide that usually undergoes an asexual cycle [1]. Its conidia are formulated as active ingredients of fungal pesticides, produced with high quality to tolerate outdoor stresses, and are of special importance for biological control. A host infection by *B. bassiana* is initiated by conidia attaching to the insect cuticle, where conidial germination leads to hyphal extension for penetration through the cuticular layers via the expression of a variety of hydrolytic enzymes and other factors [2,3,4,5]. After entry into the host hemocoel, the hyphal development turns into a process for forming plenty of blastospores that propagate rapidly until the host is mummified to death [5,6,7]. The blastospores bind to the integument and grow as hyphae that penetrate the host cuticle again to form aerial conidia to begin a new infection cycle [8,9,10]. There are two processes of dimorphic transition during infection: conidial germination leads to hyphal penetration after conidial adhesion, and then the penetrating hyphae from the host hemocoel turn into blastospores. Although the mechanism of the dimorphic switch is critical for the virulence and asexual cycle of *B. bassiana*, much remains obscure.

The asexual cycle of *B. bassiana* comprises two distinct phases, hyphal growth and conidiation, which are precisely timed and genetically programmed in response to stimulation by internal and external signals [11,12,13,14]. Conidiation is genetically controlled by a central developmental pathway (CDP) consisting of the developmental regulators BrlA, AbaA, and WetA, which mediate the development of conidiophores and the formation of conidia in *Aspergillus* [15,16]. BrlA is a key activator that initiates conidiation, followed by the sequential activation of AbaA and WetA, respectively, in the middle and late phases [17,18,19,20]. PdbrlA, PdabaA, and PdwetA of *Penicillium digitatum* were also confirmed to control distinct stages of conidiogenesis [21]. *B. bassiana* needs to survive independently in the environment before infecting fungi, so the utilization of environmental nutrients, response to external stress, and conidiation will directly affect the vitality of fungi and their ability to infect insects [1,2,3,4]. Therefore, it is also very important whether asexual development-related genes are involved in responding to external environmental stimuli. In *B. bassiana*, conidiation is completely abolished in the absence of *brlA* or *abaA* [13] and is nearly abolished in the absence of *wetA* [11], and virulence is greatly attenuated in mutants with these deletions. Asexual development is governed by BrlA, AbaA, and WetA in the central pathway of *B. bassiana.* However, the response to environmental stress has not been detected after the deletion of *brlA* or *abaA* in *B. bassiana*.

The upstream developmental activation (UDA) pathway containing the signal transducer genes *fluG* and *flbA–flbE* could initiate conidiophore development by activating the CDP genes in *Aspergillus* [15,16,22]. The FluG protein was first discovered in *Aspergillus niger*, and has since been identified in other *Aspergillus* and *Penicillium* fungi [22,23,24,25]. In *Aspergillus niger* [19], FluG mainly activates a series of activators in the UDA pathway to positively regulate the CDP genes (*brlA*, *abaA* and *wetA*) in a non-linear transcriptional regulatory cascade mode, thereby terminating the nutritional growth and initiating the asexual sporulation process. The deficiency of FluG can inhibit the sporulation and petiole, while causing defects in conidiation structure leading to the fluffy phenotype [25,26]. In *Aspergillus fumigatus,* the *fluG* knockout strain showed similar growth morphology and normal sporulation compared to the wild strain on solid culture medium, but sporulation was absent in liquid culture, indicating that FluG only plays a special regulatory role in the sporulation process of *A. fumigatus* under liquid conditions [27,28]. The absence of *fluG* also did not lead to a decrease in fluffy colony morphology and sporulation in *Penicillium digitatum* [21]. The above research results indicate that the regulatory mechanisms of FluG in the asexual sporulation process of filamentous fungi are different and have species specificity.

In particular, *fluG* is the key genetic switch that terminates the growth of hyphae and initiates conidiation by activating *flbA*–*flbE* in different ways after receiving the stimulation signal [29,30,31,32,33,34,35,36,37]. Meanwhile, *fluG* could be activated by inhibiting *sfgA*, which is a key suppressor participating in hyphal growth [24,25]. Transcription of the *flb* and CDP genes was still activated in the absence of *fluG,* and the changes were correlated with the yield changes of the conidia, implying that *fluG* is a core, but not unique, player in the UDA pathway [14]. As previously reported, *fluG* disruption in *B. bassiana* not only affects the conidial yield but also attenuates cell integrity and virulence and increases cellular sensitivities to different stresses. In view of the important role of *fluG* in the growth and development of *B. bassiana*, we comprehensively compared the differences between the upstream developmental regulatory gene *fluG* and the key genes *brlA*, *abaA*, and *wetA* in the central growth and development pathway in morphology, conidiation, nutrient utilization, the external stress response, and virulence and summarized the particular function of *fluG* located upstream of growth and development. Despite an in-depth and comprehensive analysis of the conidiation and environmental stresses of *B. bassiana*, the mechanism and signaling pathway of *fluG* remain poorly understood. This study sought to characterize the different roles of the *fluG* and CDP genes in morphology, conidiation, nutrient utilization, the external stress response, and virulence, including the signaling pathways they participate in and the downstream proteins they directly activate, in order to deepen our understanding of the mechanism of the upstream developmental regulatory gene *fluG* in growth and development, resistance to environmental stress, and virulence. 

## 2. Results

### 2.1. Comparison of fluG Disruption with brlA, abaA, and wetA Disruption of Phenotypic and Asexual Cycle 

To explore the unique role of *fluG* (BBA_04942) in the growth and development of *B. bassiana*, disruption mutants and corresponding complementation mutants of *fluG* and the CDP-related genes *brlA* (BBA_07544), *abaA* (BBA_00300), and *wetA* (BBA_06126) were constructed, as described above. All mutants were verified through PCR, RT-PCR, and qRT-PCR with paired primers (Appendix A). 

Compared with the control (wild and complemented) strains, the Δ*fluG* mutant grew slightly slower but without an apparently fluffy phenotype under the optimal regime (Figure 1A). Although the colonies of the Δ*brlA*, Δ*abaA*, and Δ*wetA* mutants had sizes similar to the control strains, they were more cottony and thicker, especially the colonies of Δ*brlA* and Δ*abaA* with greater mycelial density. Meanwhile, Δ*wetA* not only exhibited an obvious fluffy phenotype, but also had grooves on the back and heavy pigment accumulation. 

The quantification of conidiation over 7 days on SDAY plates demonstrated the different degrees of reduction in conidiation yield in each deletion mutant (Figure 1B). Compared with the control strains, aerial conidiation was totally abolished in the Δ*brlA* and Δ*abaA* mutants during the 7 days, while the Δ*fluG* and Δ*wetA* mutants suffered conidial yield losses of 73.9% and 97.1% on day 3, 90.0–97.0% and 98.3–99.7% on days 4–6, and 84.7% and 97.9% on day 7, respectively. The blastospore production level of each strain was monitored daily during an 8-day incubation in SDB culture. The Δ*brlA* and Δ*abaA* mutants completely lost the ability to produce blastospores. The Δ*fluG* mutant showed 78.5–86.4% reductions on days 3–7, and the reduction diminished to 65.7% on day 8. The Δ*wetA* mutant showed only a 15.1% reduction on day 3. Subsequently, the reduction increased to 55.9–65.7% on days 4–8. 

Aerial conidiation and blastospore production were restored by each gene complementation. The Δ*brlA* and Δ*abaA* mutants completely lost the ability to produce aerial conidia on a plate or blastospores in a bath. The Δ*wetA* and Δ*fluG* mutants suffered severe defects in conidiation, and the Δ*wetA* mutant showed slightly more defects than the Δ*fluG* mutant.

### 2.2. Different Effects of FluG, BrlA, AbaA, and WetA on Multiple Stress Responses and Virulence 

The deletion of the *fluG*, *brlA*, *abaA,* and *wetA* genes had different effects on the sensitivity of *B. bassiana* to external stress during 7 days of colony growth on CZA plates supplemented with chemical stressors (Figure 2). Compared with the WT strain, the EC_50_ values of the Δ*fluG* mutant for oxidative stress induced by menadione or H_2_O_2_ were reduced by 23.9% or 40.4%, respectively (Figure 2A,B). The Δ*wetA* mutant was also significantly more sensitive to menadione or H_2_O_2_, and its EC_50_ values were reduced by 13.6% or 7.6%, respectively (Tukey’s HSD, *p* = 0). The deletion of *brlA* and *abaA* did not cause a loss of fungal resistance to oxidative stress. The tolerance to the cell wall interference stress of Congo red or SDS decreased significantly by 40.6% or 26.7% in the Δ*fluG* mutant and by 24.8% or 32.1% in the Δ*wetA* mutant, respectively (Figure 2C,D). The Δ*brlA* and Δ*abaA* mutants also did not have decreased resistance to cell wall interference stress. The resistance to NaCl hypertonic stress or carbendazim decreased significantly, by 26.0% or 68.0% in the Δ*fluG* mutant and by 8.2% or 12.1% in the Δ*wetA* mutant, while the resistance of the Δ*brlA* and Δ*abaA* mutants did not change significantly (Figure 2E,F). 

Since the Δ*brlA* and Δ*abaA* mutants produced neither aerial conidia nor submerged blastospores, a blastospore-removed hyphal suspension of each strain was applied for normal infection through cuticular penetration by immersing (Figure 2G). In contrast to the WT strain, the semi-lethal times of the Δ*brlA* and Δ*abaA* mutants increased by 1.86 times and 2.17 times, while those of the Δ*fluG* and Δ*wetA* mutants increased by 69.6% and 48.4%, respectively.

### 2.3. Different Effects of FluG, BrlA, AbaA, and WetA on Nutrient Utilization 

Compared with the WT strain, all deletion mutants showed different degrees of growth defects in the CZA or CZA-derived media. After 7 days of standard incubation on the CZA, CZA-C, CZA-N, or CZA-C-N media, the colony sizes of the Δ*fluG* strain were significantly reduced to 75.5%, 74.5%, 77.2%, or 75.5% of those of the WT strain, respectively, while the deletion of *brlA* and *wetA* had little effect on growth (Figure 3A). When cultured on the CZA, CZA-C, or CZA-N media, the colony areas of the Δ*abaA* mutant increased significantly to 1.3, 1.3, or 1.4 times those of the WT strain, respectively. When the sole nitrogen source of CZA was replaced with 0.3% NH_4_Cl, NaNO_2_, or NH_4_NO_3_, among the mutant strains the Δ*fluG* mutant had the most significant damage due to the utilization of the substituted nitrogen source, and the colony sizes were reduced to 87.8%, 74.5%, or 92.4% of those of the WT strain (Figure 3B). When replacing the sole carbon source with 3% olive oil, maltose, trehalose, glycerol, glucose, fructose, mannitol, sorbitol, lactose, acetate, or ethanol, the Δ*fluG* mutant still had the most significant damage due to the utilization of the replaced carbon source among the mutated strains, and the colony area decreased by 13.9% to 41.7% (Figure 3C). These data implied that compared with *brlA*, *abaA,* and *wetA*, significant growth defects were observed on the CZA and CZA-derived media after the deletion of *fluG.*

### 2.4. Regulatory Roles of fluG in Global Gene Expression 

The regulatory roles of the upstream developmental regulatory gene *fluG* were examined by analyzing the transcriptomes of the Δ*fluG* and WT strains. Three replicates were derived from 3-day SDAY cultures in which conidiation was rapidly developing. Compared with the WT strain, the Δ*fluG* mutant had 1281 upregulated (~12.4% of the genome) and 1257 downregulated (~12.2% of the genome) genes (Figure 4A). The gene ontology (GO) analysis (Figure 4B) revealed that these differentially expressed genes (DEGs) were enriched for three GO terms that were involved in biological processes, cellular components, and molecular function. There were 20 GO terms for biological processes, and among them the DEGs were mainly focused on cellular processes, metabolic processes, localization, and responses to stimuli. For eight enriched cellular component terms, the DEGs mainly participated in organelles, membranes, and extracellular regions. For nine GO terms related to molecular function, the DEGs were mainly enriched for binding, catalytic activity, and transporter activity. 

In the Δ*fluG* mutant, the DEGs for the top 20 significantly enriched terms in the KEGG revealed that the upregulated genes were focused on ABC transporters, propanoate metabolism, tryptophan metabolism, and galactose metabolism, while the downregulated genes were concentrated in DNA replication, mismatch repair, and fatty acid biosynthesis (Figure 4C). The gene expression information of the ABC transporter, DNA replication, and fatty acid metabolism pathway on which the DEGs were significantly enriched and five other key signaling pathways were analyzed (Figure 4D). The results in Figure 5 revealed that the transcript levels of 22 DEGs involved in DNA replication and 8 DEGs involved in fatty acid biosynthesis were all downregulated, while the transcript levels of 17 DEGs affecting ABC transporters were all upregulated. All four DEGs involved in oxidative phosphorylation were downregulated, and all five DEGs involved in autophagy were upregulated. For the ubiquinone term, the transcript levels of 3-dehydroshikimate dehydratase (BBA_01589) and 4-hydroxyphenylpyruvate dioxygenase (BBA_08551) were upregulated to 5.92 and 4.23 times the normal levels, while 4-hydroxybenzoate polyprenyltransferase (BBA_09001) and flavin prenyltransferase (BBA_04668) were downregulated to 0.031 and 0.155 times the normal levels, which affected the normal development and metabolism of the fungi [38,39,40,41]. Four DEGs were upregulated and two were downregulated in the peroxisome pathway, which affected the ability of the fungi to resist oxidative stress. The age–rage signaling pathway is very important for maintaining cell carbohydrate and protein homeostasis. In this pathway, the expression level of NADPH (BBA_07926) was downregulated to 0.325 times the normal level, and two 1-phosphatidylinositol 4,5-diphosphate phosphodiesterases (BBA_03011 and BBA_06798) were upregulated by two-fold [42,43]. 

In order to verify the effectiveness of the transcriptome sequencing, six upregulated genes and six downregulated genes of *B. bassiana* were randomly selected for qRT-PCR analysis (Appendix A). The results showed that the relative expression trend of the 12 genes was consistent with the results of the transcriptome sequencing, which verified the validity of the results of the transcriptome analysis.

### 2.5. Screening and Analysis of Interacting Proteins 

To investigate FluG-interacting proteins, we performed a yeast two-hybrid (Y2H) screening assay. In total, 53 positive clones were screened (Appendix A). Through library screening, positive clone sequence analysis, and comparison online, 40 proteins were directly affected by FluG in the 53 positive clones. The relevant information shown in Table 1 revealed that these proteins participated in various signaling pathways such as metabolism, oxidative stress, and cellular homeostasis. Through GO enrichment analysis, the results showed that the 40 proteins were mainly concentrated in metabolic and cellular processes in the biological process term; in cells, cell components, and organelle components in the cellular component term; and in binding and catalytic activities in the molecular function term (Figure 6A). Among them, 26 proteins were significantly enriched in six signaling pathways, according to the KEGG enrichment analysis. These pathways were translation, amino acid metabolism, energy metabolism, carbohydrate metabolism, the metabolism of cofactors and vitamins, and the biosynthesis of other secondary metabolites (Figure 6B). The library screening was verified by yeast two-hybridization between five randomly selected interacting proteins and FluG (Appendix A). 

## 3. Discussion

In the upstream developmental activation pathway of aspergilli, *fluG* is required for the commencement of conidiation. It collaborates with different *flb* genes [15,16,44,45], then three sequentially active genes (*brlA*, *abaA*, and *wetA*) in the central developmental pathway are activated to mediate the development of conidiophores and conidia [17,20]. In *B. bassiana*, disruption of *brlA* and *abaA,* as described before, led to abolished conidiation and blastospore production and greatly attenuated virulence but had no negative impact on hyphal growth in various media [13]. Deletion of *wetA* and *fluG* led to reductions in conidiation of varying degrees, attenuated conidial virulence, and several defects in response to nutritional and abiotic stresses [11,14]. Since the responses of *brlA* and *abaA* deletion mutants to environmental stress had not been detected and the functions of genes located in the central developmental pathway and upstream developmental activation pathway had not been compared together, we compared the differences in morphology, conidiation, nutrient utilization, stress resistance, and virulence among mutants of these four genes (*fluG*, *brlA*, *abaA*, and *wetA*) and conducted in-depth research on the downstream regulatory mechanisms of *fluG*, as discussed below. 

In the comparison of morphology, conidiation, and blastospore production, it was found that the genes in the CDP had stronger control over conidiation and blastospore production, and the mutants had a more cottony and thicker phenotype in *B. bassiana*. Disruption of *fluG* had a limited impact on conidiation and blastospore production and had little effect on radial growth. These results indicate that *brlA* and *abaA* in the CDP served as master regulators of asexual development, while the ability of *wetA,* located downstream, to regulate asexual development was stronger than that of *fluG* but weaker than that of *brlA* and *abaA*. Both *brlA* and *abaA* have highly conserved roles in the regulation of asexual development in other filamentous fungi [20,26,46,47,48]. WetA is also a crucial regulator of conidiation capacity in *B. bassiana* but exerts negative feedback control over conidiation in a way very different from *Aspergillus fumigatus* and *Fusarium graminearum* [11,20,49]. *fluG* has a strong ability to control conidiation, but it is always inferior to the regulatory genes located in the CDP. This indicates that the *fluG*-mediated regulatory pathway is not the only way that the UDA pathway regulates asexual development.

Apart from the severe defects in conidiation, the conidia of the Δ*wetA* and Δ*fluG* mutants led to different degrees of cell wall damage, slower germination, rapid viability loss, attenuated virulence, and reduced stress tolerance in *B. bassiana* [11,14]. Due to the conidiation defects of the Δ*brlA* and Δ*abaA* mutants, experiments on conidial viability could not be carried out. In our study, we compared the phenotypic differences of the Δ*brlA*, Δ*abaA*, Δ*wetA*, and Δ*fluG* mutants in nutrient utilization, environmental stress, and virulence of hyphae to find the differences between the UDA and CDP genes. The results confirm that *brlA* and *abaA* do not participate in nutrient utilization or responses to environmental stress, but *wetA* and *fluG* play important roles, especially *fluG*. The results indicate that compared with the genes in the CDP, *fluG* is more involved in sensing external stimuli. The fungal virulence was largely attenuated via hyphal infection in the Δ*brlA* and Δ*abaA* mutants, and it was already confirmed that the difficulty of hyphal infection of an insect through cuticular penetration could be attributed to the markedly reduced transcripts of multiple genes critical for host adhesion and cuticle degradation [14]. In addition, the Δ*fluG* mutant was much more virulent than the Δ*wetA* mutant*,* more sensitive to environmental stress, and less capable of utilizing environmental carbon and nitrogen sources for growth. Different defects in nutrient utilization, responses to environmental stress, and hyphal virulence between the four genes indicate that *fluG* plays an important role in the fungus sensing stimuli from the host insect and the environment.

Here, a simple genetic model diagram for upstream and central regulators was drawn based on previous research on conidiation in *B. bassiana* (Figure 7). *brlA*, *abaA* and *wetA*, as key regulators in CDP, directly affect conidiation [11,13]. In the *flb* genes, *flbA* showed a much greater role than *flbC* in fungal conidiation, blastospore production and insect–pathogenic lifecycle. *flbB*, *flbD* and *flbE* were not influential on the expression of *brlA* or *abaA* irrespective of limited or little contribution to conidiation [50]. Therefore, the interaction relationship between *flb* genes and the mode of action on regulators in CDP are not yet clear, and further research is still needed. In a previous study, the relative transcriptional levels of all *flb* and CDP genes were consistently active in the Δ*fluG* mutant during continuous cultivation and were correlated with the yield changes of conidia and blastospores in *B. bassiana*, but were very different from those of other filamentous fungi [14,26,37,51,52]. This implied that *fluG* did not orchestrate the *flb* genes for the activation of *brlA* to facilitate conidiation.

Hence, transcriptome analysis and yeast library screening were used to further verify the signaling pathways and interacting proteins directly affected by FluG. In the absence of *fluG*, DNA replication, fatty acid biosynthesis, and oxidative phosphorylation were adversely affected, while ABC transporters and autophagy were promoted. In addition, some genes regulating normal development and metabolism, the response to oxidative stress, and cell carbohydrate and protein homeostasis were affected to varying degrees. Moreover, 40 proteins directly affected by FluG also mainly participated in metabolism, oxidative stress, and cellular homeostasis, which was consistent with the effects on their transcription levels. Although two-hybrid systems offer numerous advantages for the identification of novel protein–protein interactions and have been widely applied in the research of protein function, there are still some uncertainties [53,54]. Therefore, they provide us with reliable information on proteins interacting with FluG, but still require extensive research to determine the interaction patterns between FluG and downstream proteins.

In addition, we also found that the *flb* and CDP genes did not exist in the FluG-interacting proteins. All phenotypic and transcriptional changes in the Δ*fluG* mutants and proteins that interact with FluG imply that *fluG* did not directly interact with the *flb* genes or CDP genes to initiate conidiation. We speculated that there may be some factors regulating the *flb* genes or CDP genes in the proteins interacting with FluG, or there may be some other *fluG*-like gene(s) acting as a regulator(s) in the UDA pathway. 

## 4. Materials and Methods

### 4.1. Microbial Cultivation

The wild-type ARSEF 2860 strain of *B. bassiana* and its mutants were grown in SDAY (4% glucose, 1% peptone, 1% yeast extract, and 1.5% agar) at 25 °C with a 12:12 h light/dark cycle for hyphal growth and conidiation. The stress responses were assayed in CZA (3% sucrose, 0.3% NaNO_3_, 0.1% K_2_HPO_3_, 0.05% KCl, 0.05% MgSO_4_, 0.001% FeSO_4_, and 1.5% agar) as a control and in CZA with different stresses. *Escherichia coli* DH5α and Top 10 (Invitrogen, Shanghai, China) were cultured in LB medium plus 100 μg mL^−1^ kanamycin or 50 μg mL^−1^ ampicillin, depending upon the resistance marker, at 37 °C for plasmid cloning and propagation. *Agrobacterium tumefaciens* AGL-1 for fungal transformation was cultivated at 28 °C in YEB medium [11].

### 4.2. Generation and Identification of fluG, brlA, abaA, and wetA Mutants

The genes *fluG*, *brlA*, *abaA,* and *wetA* (tag loci: BBA_04942, BBA_07544, BBA_00300, and BBA_06126) were deleted from the WT strain via homologous recombination of their 5′ (up) and 3′ (down) coding fragments separated by the bar marker in the vector p0380-5′x-bar-3′x (x = *fluG*, *brlA*, *abaA,* or *wetA*). For targeted gene complementation, a full-length coding sequence with flanking regions was inserted into the vector p0380-sur-x. All the 5′ and 3′ fragments and full-length sequences were cloned from the WT strain with paired primers up-F/R, down-F/R and fl-F/R, and digested with appropriate restriction enzyme sites (Appendix A). All the knockout plasmids and complement plasmids for *fluG* and *wetA* were integrated into corresponding strains via *Agrobacterium*-mediated transformation [11,14]. Since the Δ*brlA* and Δ*abaA* mutants were completely unable to produce conidia, their resultant complement plasmids were ectopically integrated into protoplasts of each deletion mutant via polyethylene glycol-mediated transformation [13]. Putative deletion or complementary mutants were screened in terms of Bar resistance to phosphinothricin (200 μg mL^−1^) or Sur resistance to chlorimuron ethyl (10 μg mL^−1^), and then identified via PCR accompanied by sequencing with paired primers id-F/R.

The temporal transcript patterns of the genes in the wild-type, deletion, and complementary mutants were assessed during 3 days of growth at 25 °C with a 12:12 h light/dark cycle on cellophane overlaid on SDAY plates. Hyphal suspensions of each strain (100 μL aliquots) were spread on each plate to initiate the cultures. Total RNA was extracted separately using TRIzolTM Plus Reagent (Takara, Kusatsu, Japan) and treated with DNase I (Takara, Kusatsu, Japan) following the manufacturer’s instructions. Every 5 μg RNA sample was reverse-transcribed with a PrimeScriptTM RT reagent kit (Takara, Kusatsu, Japan). The temporal transcript patterns and relative transcript levels of the genes (18S rRNA was used as an internal standard) were assessed via triplicate RT-PCR and qRT-PCR assays with the primers listed in Appendix A. Positive deletion mutants were analyzed in parallel with the WT strain and rescued mutants as control strains in the following experiments with three independent replicates. 

### 4.3. Assays for Radial Growth, Conidiation, Blastospore Production, Hyphal Stress Responses, and Virulence

The conidia yield on SDAY were quantified as described previously [11]. Briefly, 100 μL aliquots of hyphal suspensions for each strain were spread on SDAY plates and incubated at 25 °C with a 12:12 h light/dark cycle for 7 days. From the third day of incubation onward, the conidia on the 5 mm diameter disks were washed daily into 1 mL of 0.02% Tween-80. After supersonic vibration, three samples were measured with a hemocytometer to assess the conidial concentration converted to the number of conidia cm^−2^ colony.

To examine the blastospore production, 100 μL aliquots of hyphal suspensions for each strain were inoculated in SDB broth and cultured for 7 days at 25 °C at 150 rpm. From the second day, the blastospore concentration in the broth was quantified via microscopy, and the blastospore yield was indicated as the number of cells per milliliter of broth. 

Hyphal suspensions of each strain (100 μL aliquots) were spread on SDAY plates and incubated at 25 °C with a 12:12 h light/dark cycle. Hyphal mass plugs (5 mm diameter) were bored from cultures cultivated for 3 days and attached centrally to plates of SDAY; CZA alone (as control); and amended CZA supplemented with menadione (1~4 mM) or H_2_O_2_ (15~60 mM) for oxidative stress, NaCl (0.5~2 M) for osmotic stress, and Congo red (0.5~2 mg mL^−1^) for cell wall stress. After 10 days at 25 °C with a 12:12 h light/dark cycle, the diameters of all colonies were measured as indices of their radial growth rates using two measurements taken perpendicular to each other across their centers. Typical colonies of each strain were photographed. The ratio of the colony size under a given stress compared to that in the control condition was defined as the survival index (Is). For each of the tested strains, Is = 1/[1 + exp(a + bx)], where x is the concentration (C) of each stressful chemical. When Is = 0.5, the fitted equations gave the solutions (−a/b) for the effective concentrations for the stressful chemicals to suppress colony growth by 50% (EC_50_) [11,55]. 

The virulence of each strain was detected via hyphae as described previously [13]. Since the Δ*brlA* and Δ*abaA* strains produced neither conidia nor blastospores, the fresh hyphae without blastospores in 3-day-old SDB cultures of each strain were suspended in 0.02% Tween-80 and standardized to a concentration of 10 mg mL^−1^. Three replicates of 40 larvae were separately immersed for 15 s in 40 mL aliquots of each strain. The same volume of 0.02% Tween-80 was used as a control. All treated samples were maintained at 25 °C with a 12:12 h light/dark cycle, and the survival records were monitored every 24 h until the records no longer changed for two consecutive days. LT_50_ (no. of days) was generated as an index of the fungal virulence via probit analysis of each time–mortality trend.

### 4.4. Assays for Relative Growth Areas on Different Media

Hyphal mass plugs (5 mm diameter) were bored from the cultures cultivated for 3 days on SDAY plates at 25 °C that were described above and attached centrally to the plates of CZA (3% sucrose, 0.3% NaNO_3_, 0.1% K_2_HPO_4_, 0.05% KCl, 0.05% MgSO_4_, and 0.001% FeSO_4_, plus 1.5% agar) and CZA-derived media with altered carbon/nitrogen sources. The CZA-derived media were prepared by removing the 3% sucrose, 0.3% NaNO_3_, or both from the CZA; replacing the sole carbon source with 3% olive oil, maltose, trehalose, glycerol, glucose, fructose, mannitol, sorbitol, lactose, acetate, or ethanol; and replacing the sole nitrogen source with 0.3% NH_4_Cl, NaNO_2_, or NH_4_NO_3_. After 7 days of incubation at 25 °C with a 12:12 h light/dark cycle, the diameter of each colony was cross-measured to calculate the growth area. The relative growth area was determined as the ratio of the mutant strain’s growth area to the wild strain’s growth area.

### 4.5. RNA Extraction, cDNA Library Construction, and RNA-Seq

The total RNA of the wild strain and the Δ*fluG* mutant was extracted from the cultures on SDAY plates after 3 days according to method 4.2. The total high-quality RNA isolated from three independent biological replicates of the samples was used to construct independent cDNA libraries. 

The RNA-Seq of two samples, namely the wild strain (control) group and the Δ*fluG* group, each with three biological replicates, was performed on an Illumina Hiseq 2000 platform, from which 150 bp paired-end reads were generated. Raw sequences were deposited in the NCBI Short Read Archive (SRA) database (http://www.ncbi.nlm.nih.gov/Traces/sra/, accessed on 21 February 2024). The accession number of the RNA-Seq data was PRJNA1078435. Raw reads in the FASTQ format were first filtered by removing any reads containing adapter sequences and low-quality reads. At the same time, the Q20, Q30, GC content, and sequence duplication level of the clean data were each calculated. The cleaned reads were mapped onto the *B. bassiana* reference genome [56] using the Bowtie2 V2.5.2 software tool [57]. 

### 4.6. Transcriptome Annotation, Expression Profiling, Data Analysis, and Data Validation

Gene function was annotated based on homology searches in the NCBI non-redundant protein (Nr), NCBI nucleotide (Nt), Swiss-Prot protein, euKaryotic Orthologous Groups (KOG), Kyoto Encyclopedia of Genes and Genomes (KEGG), Gene Ontology (GO), and Protein family (Pfam) databases.

The expression levels of genes were measured using counts of reads normalized based on their respective lengths in the Cufflinks 2.0.2 package using its default settings (http://cole-trap-nell-lab.github.io/cufflinks/, accessed on 2 November 2023) for normalization (genometric), followed by a distribution analysis in terms of Fragments Per Kilobase of exon per Million mapped reads (FPKM) units. The differentially expressed genes (DEGs) seq package was used to identify DEGs between the control and gene mutant groups; it provided statistical routines for determining differential expression levels in the digital gene expression data by applying a model based on the negative binomial distribution. The resulting *p*-values were adjusted using Benjamini and Hochberg’s approach to control the false discovery rate (FDR). Genes with an adjusted *p*-value < 0.05, as detected by DEGSeq, were designated as differentially expressed. The fold-change in a given gene’s expression between samples was calculated as log2 (treatment FPKM value/control FPKM value).

The DEG-related signaling pathways were analyzed in *B. bassiana* using the KEGG [58,59,60]. Related maps were also obtained from the KEGG. 

To confirm the results of the transcriptome comparisons, qRT-PCR was performed on 16 randomly selected genes in *B. bassiana* [61]. The samples were consistent with those used by the transcriptome. The primers for qRT-PCR are listed in Appendix A. 

### 4.7. Nuclear cDNA Library Construction and Yeast Two-Hybrid Library Screening

For nuclear library construction, RNA was isolated from the wild strain after inoculation on SDAY for 3 days. A nuclear library of 1.12 × 10^7^ mL^−1^ clones was constructed with the vector pGADT7 (Takara Bio, Japan) according to the CloneMiner instructions by OEBiotech (Shanghai, China). For library screening, an open reading frame with *fluG* as the bait construct was generated with the vector pGBKT7 (Takara Bio, Japan). Yeast two-hybrid (Y2H) screening was performed using a Matchmaker Two-Hybrid system (Clontech). Briefly, the bait and cDNA library plasmids were co-transformed into Y2HGold competent yeast cells. After transformation, the co-transformants were plated onto SD/-Trp/-Leu/X-α-Gal/AbA (DDO/X/A) agar plates. Single blue colonies were selected and plated onto higher-stringency SD/-Trp/-Leu/-His/-Ade/X-α-Gal/AbA (QDO/X/A) plates to test reporter gene expression. Positive colonies were thereafter re-seeded in SD/-Trp/-Leu/-His/-Ade liquid media. Prey plasmids were extracted from putatively positive clones using an Easy Yeast Plasmid Isolation Kit (Clontech, Mountain View, CA, USA) and sequenced. After sequencing with the primer T7, the *B. bassiana* genes corresponding to the inserts in these clones were identified via Blast searches and then analyzed using the KEGG database. 

To further confirm the interactions, the full-length cDNA sequences of related genes were amplified and inserted into the pGADT7 vector, which was co-transformed with pGBKT7 bait plasmids into yeast strain Y2HGold and co-cultured on DDO/X/A and QDO/X/A plates to test for interactions. The same positive and negative controls used in the Y2H screening were included in these experiments.

### 4.8. Statistical Analyses

All samples from three repeated assays were quantitative indices for phenotypic changes among the tested strains in a one-way analysis of variance. Significant differences among the wild, gene disruption, and complementation mutant strains were determined with Tukey’s honest significance test (Tukey’s HSD). 

## 5. Conclusions

After comparing the regulation of morphology, resistance to external stress, virulence, and nutrient utilization capacity between the upstream developmental regulatory gene *fluG* and the key genes *brlA*, *abaA*, and *wetA* in the central growth and development pathway in *Beauveria bassiana*, we found that *fluG* serves as an important regulator in the UDA pathway and has less ability to control asexual development than genes in the CDP, but can strongly sense stimuli from host insects and the environment. FluG mainly affects fungal metabolism, oxidative stress, and cellular homeostasis, but does not directly interact with the *flb* genes or CDP genes to initiate conidiation. These results led to speculation that warrants further research on the proteins interacting with FluG and other *fluG*-like genes existing in the UDA pathway of asexual development.

## Figures and Tables

**Figure 1 ijms-25-06261-f001:**
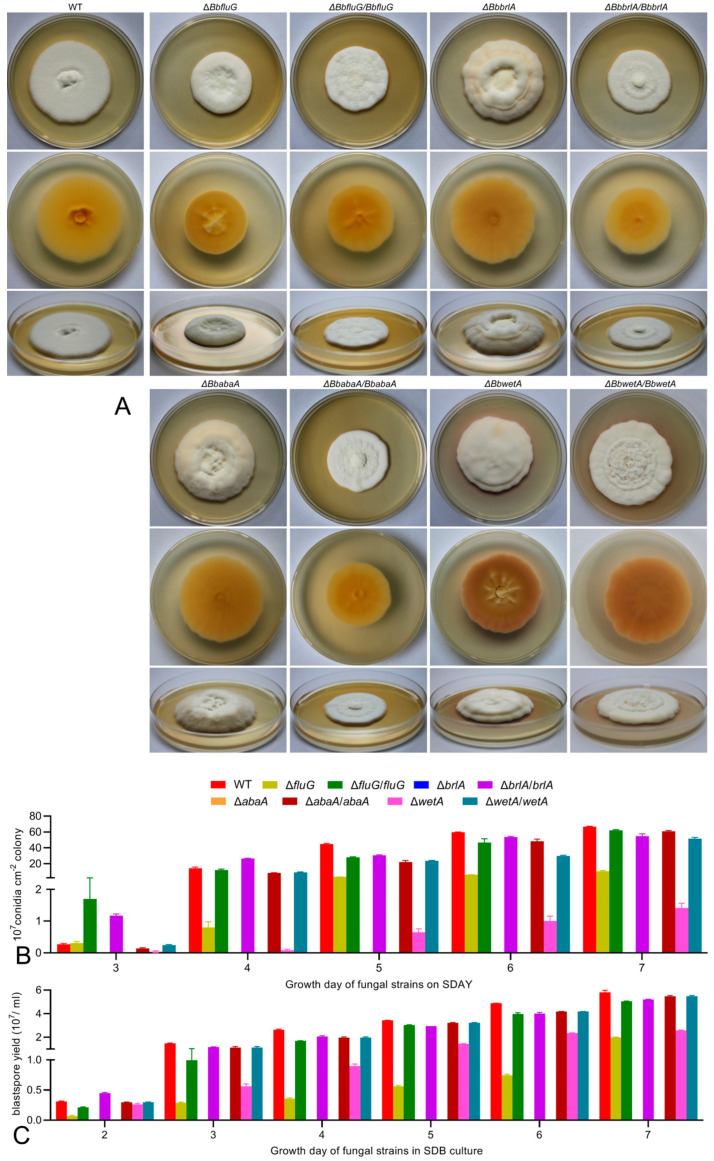
Effects of *fluG*, *brlA*, *abaA,* and *wetA* on vegetative growth, aerial conidiation, and blastospore production of *B. bassiana*. (**A**) Top (row 1), bottom (row 2), and side (row 3) views of fungal colonies initiated with hyphal mass plugs (5 mm diameter) and cultivated for 10 days on SDAY at 25 °C. (**B**) Conidial yields from SDAY cultures initiated with 100 μL of hyphal suspension per plate and grown for 7 days at 25 °C with a 12:12 h light/dark cycle. (**C**) Blastospore yields in the submerged SDB cultures over 7 days of incubation at 25 °C and 150 rpm.

**Figure 2 ijms-25-06261-f002:**
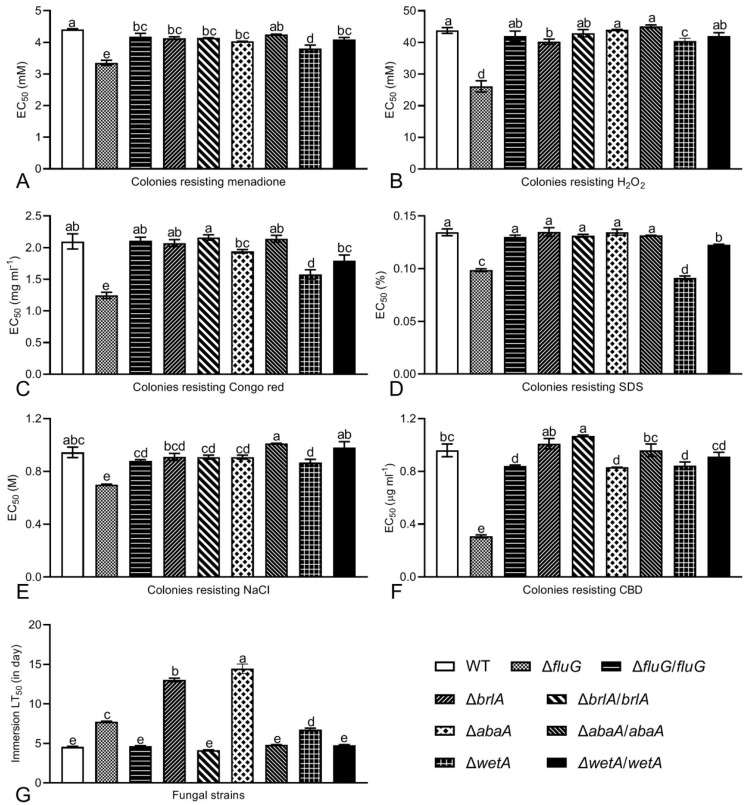
Contributions of *fluG*, *brlA*, *abaA*, and *wetA* to multiple stress responses and virulence of *B. bassiana* during growth. (**A**–**F**): EC_50_ values of chemical stressors required to suppress radial growth by 50% after 7 days of cultivation on CZA plates at 25 °C with 12:12 h light/dark cycle. (**G**) LT_50_ (no. of days) for hyphal virulence to *G. mellonella* larvae inoculated via topical application (immersed). Note: different letters on the bars denote significant differences in each group (Tukey’s HSD, *p* < 0.05). Error bars: SDs from three replicates.

**Figure 3 ijms-25-06261-f003:**
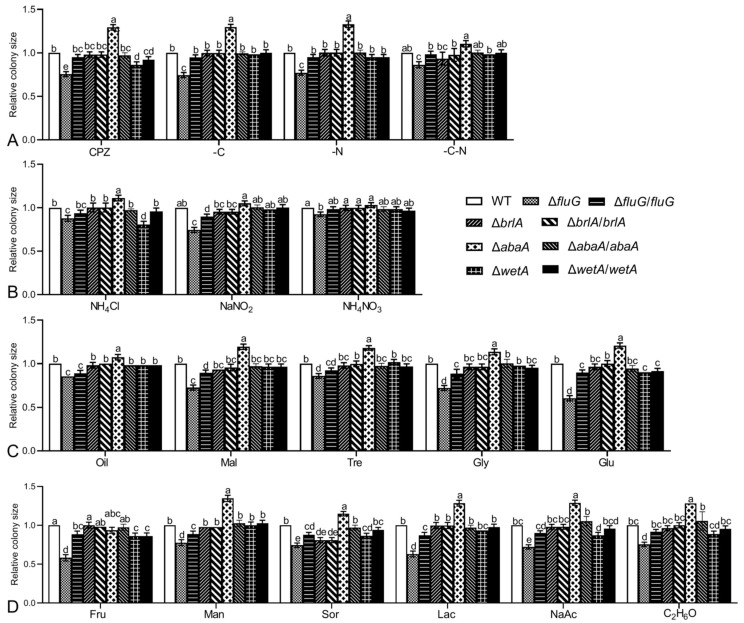
Effects of *fluG*, *brlA*, *abaA,* and *wetA* on nutrient utilization of *B. bassiana*. (**A**–**D**) Relative colony sizes of *B. bassiana* strains initiated with hyphal mass plugs with 5 mm diameters, measured as the ratio of the mutant strain’s growth area to the wild strain’s growth area after 7 days of cultivation on various media at 25 °C. Note: different letters on the bars denote significant differences in each group (Tukey’s HSD, *p* < 0.05). Error bars: SDs from three replicates.

**Figure 4 ijms-25-06261-f004:**
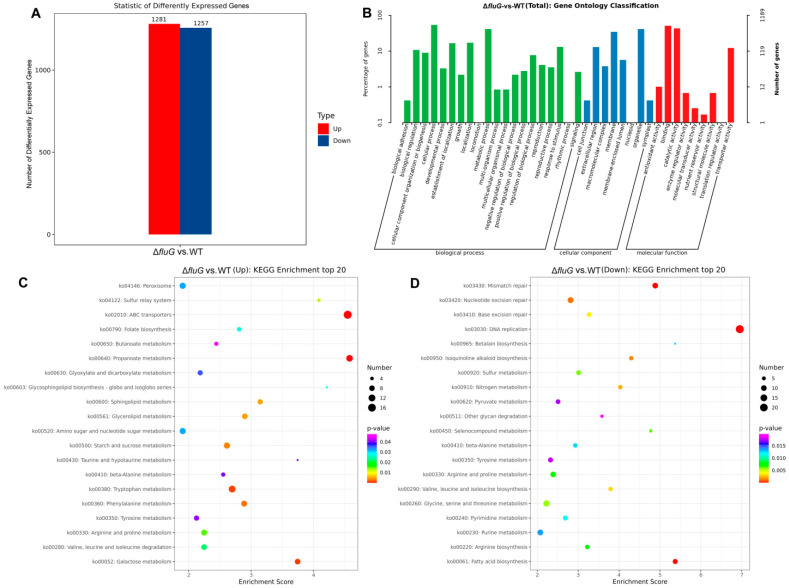
Overview of RNA-Seq data. (**A**) Histogram of differentially expressed genes (DEGs) from the Δ*fluG* mutant compared with the WT. Red and blue colored bars represent significantly upregulated and downregulated genes, respectively. (**B**) Gene ontology (GO) classification of DEGs from the Δ*fluG* mutant compared with the WT. The enriched GO terms are along the vertical axis, and the horizontal axis indicates the percentage of DEGs in a given term. (**C**,**D**) Kyoto Encyclopedia of Genes and Genomes (KEGG) pathway enrichment analysis of up- and downregulated DEGs from the Δ*fluG* mutant compared with the WT. The ordinate represents the pathway name, the abscissa represents the enrichment factor, and the point size represents the number of DEGs in that pathway, while the point colors denote the differing Q-value ranges.

**Figure 5 ijms-25-06261-f005:**
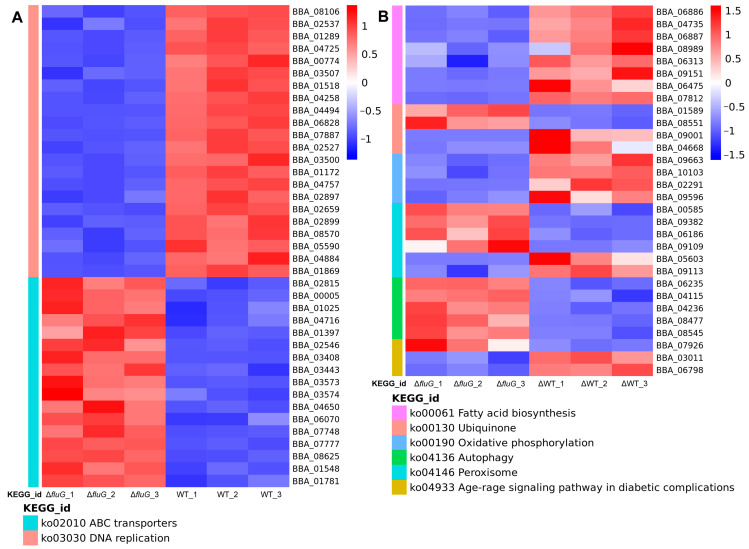
Heat map analysis of DEGs in key signaling pathways. (**A**,**B**) The relative transcript levels of DEGs in eight important signaling pathways were analyzed and are shown using a heat map.

**Figure 6 ijms-25-06261-f006:**
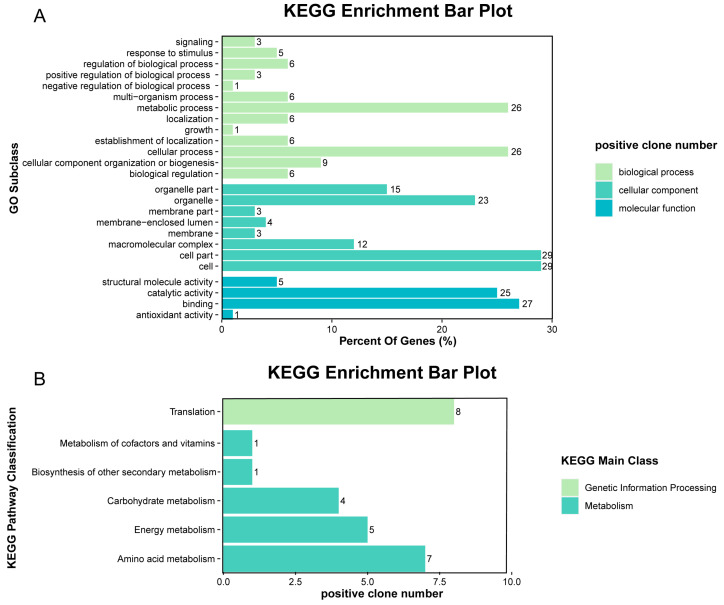
GO classification and KEGG pathway enrichment analysis of proteins that FluG interacted with. (**A**,**B**): GO classification and KEGG pathway enrichment analysis of proteins that were selected by yeast two-hybrid library screening.

**Figure 7 ijms-25-06261-f007:**
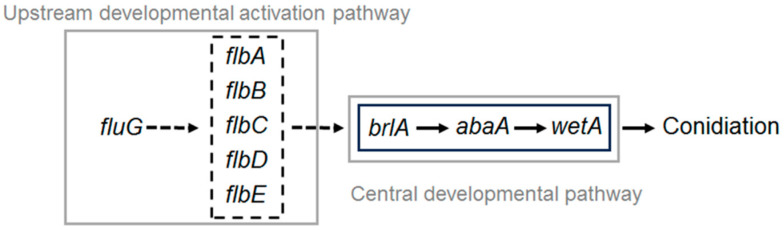
Genetic model diagram for upstream and central regulators in *B. bassiana*.

**Table 1 ijms-25-06261-t001:** Sequence analysis of screening of positive clones using *fluG* sequence as bait.

Clone No.	Gene_id	Gene_Name	Clone No.	Gene_id	Gene_Name
1	BBA_09035	phenol 2-monooxygenase	34	BBA_02100	hypothetical protein
2	BBA_05531	cytochrome b-c1 complex subunit 2	35	BBA_06761	zinc finger and BTB domain-containing protein 7A
3	BBA_10085	SWI/SNF and RSC complexes subunit ssr3	36	BBA_01481	ubiquitin carrier protein
4	BBA_03721	alkaline phosphatase	37	BBA_06393	malate dehydrogenase
5	BBA_01606	isocitrate/isopropylmalate dehydrogenase	38	BBA_04259	carbon catabolite derepressing protein kinase
6	BBA_08294	biogenesis of lysosome-related organelles complex 1 subunit KXD1	39	BBA_09482	polynucleotide kinase 3 phosphatase
7–11	BBA_08577	ribosomal protein S7	40	BBA_00422	Actin-like protein ARP9
12–17	BBA_08182	3-hydroxybenzoate 6-monooxygenase	41	BBA_01918	hypothetical protein
18	BBA_04319	transcriptional regulatory protein pro1	42	BBA_10102	cyclic-amp-dependent transcription factor atf-2
19–20	BBA_03167	glycosyl hydrolase 53 domain-containing protein	43	BBA_06969	prolyl-tRNA synthetase
21	BBA_07782	hypothetical protein	44	BBA_03016	peptidase family protein
22	BBA_03527	tRNA intron endonuclease	45	BBA_01844	tat pathway signal sequence
23–25	BBA_00726	Glutamine synthetase	46	BBA_00241	Ssu72-like protein
26–27	BBA_02314	COP9 signalosome complex subunit 5	47	BBA_08019	threonyl-tRNA synthetase
28	BBA_00241	RNA polymerase II subunit A C-terminal domain phosphatase Ssu72-like protein	48	BBA_07743	serine peptidase
29	BBA_02591	C6 zinc finger domain protein	49	BBA_06377	RasGEF domain-containing protein
30	BBA_08096	putative Zn(II)2Cys6 transcription factor	50	BBA_05170	acetamidase/formamidase family protein
31	BBA_02631	PIR protein repeat protein	51	BBA_09338	catalase/peroxidase HPI/catalase P
32	BBA_08791	alkaline serine protease AorO	52	BBA_02558	ferulic acid esterase (FaeA)
33	BBA_01621	nitrogen permease regulator 2	53	BBA_00422	Actin-like protein ARP9

## Data Availability

All data generated or analyzed during this study are included in the published paper and associated Appendix A. Raw sequences were deposited in the NCBI Short Read Archive (SRA) database (http://www.ncbi.nlm.nih.gov/Traces/sra/, accessed on 21 February 2024). The accession number of RNA-seq data was PRJNA1078435.

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
