# Peer review of "Comprehensive Insights into the Remarkable Function and Regulatory Mechanism of FluG during Asexual Development in Beauveria bassiana"

_ijms, 2024, doi:10.3390/ijms25116261_

Round 1
Reviewer 1 Report
Comments and Suggestions for Authors
In the manuscript ‘Comprehensive insights into the remarkable function and regulatory mechanism of FluG during asexual development in Beauveria bassiana’, the authors constructed four deletion mutants: DfluG, DbrlA, DabaA and DwetA in the entomopathogenic fungus Beauveria bassiana, further performed phenotypic and transcriptomic analysis, which shown that the FluG protein is relevant for spore formation. Based on the transcriptomic and two-hybrid analysis the authors concluded that FluG is involved the DNA replication, cell homeostasis, growth and stress response.
In my opinion, the topic analyzed here, may be of interest for the readers of IJMS and for fungal community. However, I have a major remark regarding this work: the entire manuscript must be rewritten, obeying the rules of English grammar. Language correction by the person fluent in English is required.
Major remarks:
Some of the presented results were already published yet (PMID: 26243054) in Springer Link publisher, in 2015y. The figure 1 contains once published pictures, construction of the DwetA strain and some growth tests. The copyrights of the publisher might be violated. Hence, the suspicion of self-plagiarism I leave to the editor's decision.
The relation between all conidiation-involved components is unclear. Schematic figure presented regulatory cascade with particular proteins would be helpful to understand it. The same in the discussion section.
Lack of information what is exactly FluG. Some functional analysis and description would be helpful.
Row 150: it is doubtful that any organism may grow on CZA medium completely without carbon nor nitrogen source.
Row 157: regarding that Beauveria represents entomopathogens, it would be expected that better carbon sources will be proteins (acquired from insects), chitin and glucosamine. What was the criterion to select these carbon sources, suitable for saprotrophic fungi? The citation needed.
The results from the Two-hybrid analysis should be also revised in comparison with laborious work of Golemis et al. (PMID: 11530601, PMID: 10683744), since many artefacts are located in the table 1 (ribosomal protein, endonuclease, polymerase, Zn-transcription factors). In the discussion section these results should be also analysed.
Row 318: where are the microelements required for fungal growth? The medium with iron source only is a very poor one and non-physiological.
Minor remarks:
In the title ‘remarkable’ word is unnecessary. FluG function is almost untouched here.
Row 123-125: confusing logical quantifier AND/OR
Figure 2G is not indicated in the text.
The two-hybrid system seems to have its own proper name and it is not nuclear system.
Row 340: isolation of RNA should be fused with the section 4.5.
Comments on the Quality of English LanguageThe entire manuscript should be verified by person skilful in the rules of English grammar. Admitting a comprehensible text lies with the editor site and is not the job of the reviewer. Reviewer is not for correction of fundamental grammatical errors.
Author Response
Dear the editor:
Please find the updated version of our manuscript (manuscript ID: ijms-3011500).
In this re-submission, we considered all comments and suggestions from the reviewers, to whom we would like to express our sincere thank. We revised the whole manuscript carefully to avoid language errors. Our one-by-one responses to the reviewers’ comments are attached below (the second page). We hope our revision and responses to be pertinent to their comments and suggestions.
Thank you very much for handling our manuscript.
Sincerely,
Fang Li

Reviewer 2 Report
Comments and Suggestions for Authors
Line 44: It is suggested to change complemented for transformants.
Line 94-97: The strain with radial growth similar to the WT was ΔbrlA, even with greater mycelial density, compared to the rest of the mutants. Consider this comment. Likewise, there are many images, just to place panel A. It would be good if the authors indicated what they are referring to in the Fig. 1. Even the image of the plates on the back does not provide much information.
Line 135: The authors indicate that their data are significantly different because they obtained P > 0.05, but it is confusing since they used P < 0.05. Here you should place the value obtained in the Anova and Tukey test.
Fig. S1. Section A is not clear enough. The authors could improve it to understand the constructs.
Fig. S1. Define in the text of the figure up-F, up-R, dn-F, dn-R, id-F and id-R. Also, indicate the full name of the promoter Ptrc
Fig S1. In the disruption of the fluG and brlA genes, when the authors performed the PCR to detect the deletion of these genes in the mutants, what size of amplicons did they expect in B and C. In addition, the authors confirmed the fragments by sequencing or only by the fragment obtained. Because the authors did not perform Southern blot to confirm the integration of the vectors, they must at least show the sequencing.
Section 4.1 Escherichia coli and Agrobacterium tumefaciens in italics.
Line 168: Note: Asterisked bars in each group differ significantly from those unmarked (Tukey’s HSD, P < 0.05). According to the figure and panels, the statistical analysis refers to the bars with (a), but it is difficult to understand at first instance. You could change to a solid color bar (white or black).
Line 215-217: The authors indicate: In order to verify the effectiveness of transcriptome sequencing, 6 up-regulated genes and 6 down-regulated genes of B. bassiana were randomly selected for qRT-PCR analysis (Figure S2). Why did they randomly select those genes to analyze their expression, if they had the transcriptome data?
Line 330: Correct the word Frangment
Comments on the Quality of English LanguageRequires moredate revision of English
Author Response

(The authors gave the same response as above.)

Round 2
Reviewer 1 Report
Comments and Suggestions for Authors
In the present form, the manuscript is more easier to read and understand. I have no other remarks
Comments on the Quality of English LanguageThe language is quite fine, enough to understand
Reviewer 2 Report
Comments and Suggestions for Authors
After reviewing the manuscript entitled: Comprehensive insights into the remarkable function and regulatory mechanism of FluG during asexual development in Beauveria bassiana. In accordance with the observations and modifications made to the paper, I suggest that the document be accepted for publication in IJMS.